# Assessment of RNA extraction protocols from cladocerans

**Muhammad Raznisyafiq Razak** [1], **Ahmad Zaharin Aris** [1,2]*, **Fatimah Md Yusoff**[2,3], **Zetty Norhana Balia Yusof**[4,5], **Sang Don Kim**[6], **Kyoung Woong Kim**[6]

**1** Faculty of Forestry and Environment, Department of Environment, Universiti Putra Malaysia, Selangor, Malaysia, **2** International Institute of Aquaculture and Aquatic Sciences, Universiti Putra Malaysia, Negeri Sembilan, Malaysia, **3** Faculty of Agriculture, Department of Aquaculture, Universiti Putra Malaysia, Selangor, Malaysia, **4** Faculty of Biotechnology and Biomolecular Science, Department of Biochemistry, Universiti Putra Malaysia, Selangor, Malaysia, **5** Institute of Bioscience, Universiti Putra Malaysia, Selangor, Malaysia, **6** School of Earth Sciences and Environmental Engineering, Gwangju Institute of Science and Technology, Buk-gu, Gwangju, Republic of Korea

* zaharin@upm.edu.my

**Data Availability Statement:** All relevant data are within the manuscript and its Supporting Information files.

**Funding:** This work was supported by GIST Research Institute (GRI) grant funded by the

## Abstract

The usage of cladocerans as non-model organisms in ecotoxicological and risk assessment studies has intensified in recent years due to their ecological importance in aquatic ecosystems. The molecular assessment such as gene expression analysis has been introduced in ecotoxicological and risk assessment to link the expression of specific genes to a biological process in the cladocerans. The validity and accuracy of gene expression analysis depends on the quantity, quality and integrity of extracted ribonucleic acid (RNA) of the sample. However, the standard methods of RNA extraction from the cladocerans are still lacking. This study evaluates the extraction of RNA from tropical freshwater cladocerans *Moina micrura* using two methods: the phenol-chloroform extraction method (QIAzol) and a column-based kit (Qiagen Micro Kit). Glycogen was introduced in both approaches to enhance the recovery of extracted RNA and the extracted RNA was characterised using spectrophotometric analysis (NanoDrop), capillary electrophoresis (Bioanalyzer). Then, the extracted RNA was analysed with reverse transcription polymerase chain reaction (RT-PCR) to validate the RNA extraction method towards downstream gene expression analysis. The results indicate that the column-based kit is most suitable for the extraction of RNA from *M. micrura*, with the quantity (RNA concentration = 26.90 ± 6.89 ng/µl), quality (A260:230 = 1.95 ± 0.15, A280:230 = 1.85 ± 0.09) and integrity (RNA integrity number, RIN = 7.20 ± 0.16). The RT-PCR analysis shows that the method successfully amplified both alpha tubulin and actin gene at 33–35 cycles (i.e. Ct = 32.64 to 33.48). The results demonstrate that the addition of glycogen is only suitable for the phenol-chloroform extraction method. RNA extraction with high and comprehensive quality control assessment will increase the accuracy and reliability of downstream gene expression, thus providing more ecotoxicological data at the molecular biological level on other freshwater zooplankton species.

Gwangju Institute of Science and Technology (GIST) in 2021.

**Competing interests:** The authors have declared that no competing interests exist.

## Introduction

Comprehensive reports on the hazards, dose-response and exposure of chemicals are necessary to establish a precise assessment of chemicals' effects on humans and the ecosystem [1]. Conventional ecotoxicological assessment is highly costly, time-consuming and dependent on testing animals as surrogates for exposure effects on humans [2]. Conventional assessments only emphasise traditional apical endpoints rather than evaluating the specific pathway of the chemicals. The pathway assessment is essential to assess the interactions of cells with tissues, tissues with organs and organs with the individual [3]. Restriction on the use of animals in ecotoxicological assessment is enforced by the European Union unless no other possible method is applicable [4,5]. The National Institutes of Health (NIH) in the United States suggested ecotoxicological testing using animal models should construct the assessment methodology using the Replacement, Reduction, and Refinement strategy [6]. Alternative computerised analysis programs to regulate emerging chemicals instead of animal models were established under the 'Replacement' strategy. The approach also encourages the usage of cells and tissues, including the substitution from higher taxonomic rank animals (mammals and primates) to lower taxonomic rank animals (invertebrates). Subsequently, the number of animals used per test would be reduced via the 'Reduction' strategy by preventing any irrelevant replication for each assessment process. Efficient and accurate experimental design through 'Refinement' would minimise test animals' suffering.

Risk assessors have begun to utilise non-model organisms such as aquatic invertebrates as bioindicators in ecotoxicological studies [7–10]. As a result, the amount of molecular assessment of zooplankton in the ecotoxicity discipline has intensified in recent years [11–18]. Among the zooplankton species, cladocerans are the group most responsive to the miniscule concentration of toxicants, as compared with copepods and rotifers [16]. Cladocerans are located in the middle of the food chain and will act as a warning alarm towards any perturbations in the aquatic ecosystem [19]. For example, cladocerans generate a vital connection between primary production (diatom-rich phytoplankton) and higher trophic organisms (fish). Inexpensive cladocerans species have reduced the cost of toxicity testing; for example, laboratory sizes can be shrunk by 50% compared to those needed for fish/animal models [20]. However, most researchers used temperate cladocerans species such as *Daphnia magna* as a species model, whereas tropical zooplankton species have not been thoroughly studied [21,22].

*Moina micrura* is a native cladoceran that is highly abundant in tropical countries including Malaysia, Thailand, the Philippines, China, Sri Lanka, India, Mexico and Brazil [23–29]. Due to their high ecological relevance, native species should be utilised in ecotoxicological studies to show the actual potential risks posed to tropical freshwater life [30]. There are fewer logistical restraints due to the high availability of this species in the wild, thus prohibiting another foreign species from being introduced into the native ecosystems [31]. Numerous study utilised *M. micrura* as a bioindicator in ecotoxicological and risk assessment due to the sensitivity of the species. For example, *M. micrura* is the most sensitive species towards cyanobacteria compared to tropical cladocerans *Daphnia laevis* and temperate cladocerans *Daphnia similis* [32]. Furthermore, *M. micrura* was the most sensitive species towards chlorpyrifos with the $LC_{50}$ value 0.08 μg/L compared with *Ceriodaphnia dubia*, *D. magna* and *Daphnia duplex* with the $LC_{50}$ value of 0.117 μg/L, 0.30 μg/L and 1.0 μg/L respectively [33]. Similar with other cladocerans, *M. micrura* possess a rapid life cycle and significant offspring due to the parthenogenesis reproduction system [34]. These characteristics permit a large number of assessments, including multigenerational experiments, with shorter time frames and minimised costs [35]. The organism's transparent carapace allows novel organ endpoints such as eggs in the brood chamber, gut and heart to indicate specific mechanistic responses [3]. Cladocerans' unique

characteristics allow the prospect of more comprehensive ecotoxicological studies at different organizational levels such as molecular, organ, individual, and population level.

Despite the advantages of cladocerans as a bioindicator in the ecotoxicity discipline, the absence of established protocols to extract nucleic acids from cladocerans have interrupted the assessment at molecular levels. Gene expression is an example of a molecular assessment used by molecular biologists and environmental toxicologists to link the expression of specific genes to a biological process [36]. This assessment can assist scientists in recognising biological pathways and identifying the genes that regulate cell behaviour, development and disease [37]. However, the consistent and comprehensive analysis of gene expression depends on the quantity, quality and integrity of extracted RNA of the sample [38]. Subsequently, RNA extraction methods can be divided into two primary categories: conventional phenol-chloroform methods and silica-based kits [39]. The methods have been modified with several reagents to enhance the quantity, quality and integrity of RNA. For example, glycogen was added during the RNA extraction process to increase the RNA quantity and quality in different types of samples such as human [40,41] and rat [42] and insects [43,44]. Glycogen contain a highly purified polysaccharide and commonly used as an inert carrier and acts as a free host to RNA [45]. Due to the insoluble characteristics of an ethanol solution, glycogen produces a precipitate that holds nucleic acids [46]. This precipitate will form a visible pallet, which will ease the handling of nucleic acids.

Most zooplankton, including cladoceran species, consists of 18S and 28S rRNA. High- integrity RNA will show distinguished 18S and 28S rRNA peaks, with no smearing towards a smaller nucleotide (nt) [43]. Agarose gels are the conventional method to assess 18S and 28S rRNA bands. However, this method has several drawbacks; for example, the rRNA bands can be influenced by the electrophoresis conditions, the volume of loaded RNA and the saturation of ethidium bromide fluorescence [47]. **Table 1** shows the summary of the study's analysis to characterise the RNA quantity, quality and integrity. As the table indicates, this study conducted the analysis using the bioanalyzer instrument, which combines fluorescence, microfluidics and capillary electrophoresis to minimise errors during the evaluation of RNA integrity. The bioanalyzer utilised automatic microfluidics-based electrophoretic techniques that determine the RNA quality value based on the analysis of digitalised electropherograms by proprietary algorithms [48]. Additionally, the minimal input as low as 50 pg/μl of RNA sample permitted the characterisation of RNA integrity [49]. This feature is critical for determining RNA integrity in small samples like cladocerans because the quantity of extracted RNA is not high.

In this study, a comprehensive assessment of RNA extracted from Cladocerans *M. micrura* with the addition of glycogen was reported for both the phenol-chloroform extraction method and the column-based kit method. This study provides a reliable and simple method for the molecular assessment of *M. micrura* for downstream applications such as reverse transcription polymerase chain reaction (RT-PCR) and next-generation sequencing (NGS) for potential application in functional genomic research such as gene expression profiling.

## Material and methods

### Chemicals and reagents

The chemicals, reagents, micropipette tips and micropestles used were molecular grade and certified Rnase-free. Chloroform, isopropanol, ethanol, β mercaptoethanol were purchased from Merck (Steinheim, Germany). RNase- QIAzol reagents, free water and Qiagen Micro kits were acquired from Qiagen (Hilden, Germany). RNA-grade glycogen was obtained from Thermo Fisher Scientific (MA, United States). To ensure a high quality of RNA extraction, workspaces such as lab benches and laminar flows were cleaned using 100% ethanol each time

**Table 1. Summary of analysis to characterise RNA quantity, quality and integrity.**

| Characterization of RNA analysis | Concentration | Purity | Integrity | Linear Range of Detection | Utilisation of Toxic Reagent | Equipment/ Supplies Required | Relative Cost per Assay | Assay Time | Hands-On Time | Ability to Automate |
|---|---|---|---|---|---|---|---|---|---|---|
| **Absorbance** | Y | Y | N | NanoDrop Spectrophotometer: 2–12,000 ng/µl | No | Spectrophotometer | Low | <1 minutes | <1 minutes | Yes |
| **Dye-Based Quantification** | Y | N | N | QuantiFlour RNA system: 0.1–500 ng Quant-iT RiboGreen RNA Assay Kit: 0.2–200 ng Qubit RNA BR Assay Kit: 20–1,000 ng | Yes | Fluorometer, Dye Kits | Medium | 15–30 minutes | 15–30 minutes | Yes |
| **Agarose and Acrylamide Gels** | Y/N[1] | N | Y | Ethidium bromide: > 34 ng SYBR Green II: >14ng SYBR Gold: >4.3 ng | Yes | Gel Box, Stain | Low | 10–120 minutes | 5–15 minutes | No |
| **Agilent 2100 Bioanalyzer** | Y | N | Y | Agilent RNA 6000 Nano Kit: 5–500 ng/µl Agilent RNA 6000 Pico Kit: 50–5,000 pg/µl | No | Bioanalyzer, Chips | High | ~1 hours | ~30 minutes | No |
| **qPCR and RT-qPCR** | Y | Y/N[2] | Y/N[3] | Depends on many factors including target, reaction conditions, RNA quality and Primer design | No | Real-Time Instrument | High | 1–2 hours | 15–30 minutes | Yes |

Y = Can be determined, N = Cannot be Determined.

[1]Qualitative, or by densitometry, quantitative.

[2]Delayed Cq values may be indicative of low purity or contamination.

[3]Delayed Cq values may be indicative of poor integrity.

[4]Depends on dye or system used.

[5]Using nucleic acid-specific dyes.

to ensure their cleanliness and to remove all contaminants. During handling of the reagent and RNA samples, gloves are utilised at all times and changed regularly as the protocol progresses from crude extract to more purified material.

## *M. micrura* culture specifications

Live samples of Cladocerans *M. micrura* were obtained from the culture in the Aquatic Animal Health and Therapeutics Laboratory (AquaHealth), Institute of Bioscience, Universiti Putra Malaysia. The culture was cultivated based on the International Organization for Standardization (ISO) method (ISO 6341:2012). The culture specifications were regulated to the light/dark cycle of 12: 12 h and $27 \pm 1°$C. The media was changed every two days with the daily supplementation of a green microalga species, *Chlorella vulgaris* ($1.0 \times 10^{6\sim8}$ cells/ ml, 1 ml/day), as the food supply.

## *M. micrura* controlled samples collection

The standardization of sample collection to ensure the M. micrura in controlled condition throughout the RNA extraction process. Live *M. micrura* females were sorted from the culture sample under a Zeiss Axioskop 2 microscope (Zeiss, Germany). Females of *M. micrura* were collected similarly from the controlled culture prior to the RNA extraction. All experiments were performed in accordance with UK legislation under the Animals (Scientific Procedures) Act 1986 Amendment Regulations (SI 2012/3039).

## RNA extraction

QIAzol (Method A).

1. Ten (10) and thirty (30) individuals of female *M. micrura* were transferred into a microcentrifuge tube.

2. Deionised water (1 ml) was added and the samples were gently rinsed using a micropipette to wash away any impurities attached to them. The samples were transferred into a new microcentrifuge tube.

3. QIAzol (100 μl) was added into the microcentrifuge tubes.

4. The samples were homogenised using a cordless microtube homogeniser (Bel-Art, NJ, USA) attached to the disposable plastic micropestle (Bel-Art, NJ, USA). The homogeniser was rotated 30 to 50 times under pressure until no large particles were visible under direct light.

5. The micropestle was rinsed with fresh QIAzol (900 μl) of reagent added into the microcentrifuge. The samples were vortexed briefly and incubated for 5 minutes at room temperature.

6. Chloroform (200 μl) was added to the samples, vortexed briefly and incubated at room temperature for 3 minutes.

7. The samples were centrifuged at >12,000 rpm, 4˚C for 15 min. The supernatant (upper layer) was transferred to a clean microcentrifuge tube.

8. Isopropanol (500 μl) was added, and then the samples were vortexed briefly and incubated for 10 minutes at room temperature.

9. The samples were centrifuged at >12,000 rpm in 4˚C for 20 minutes and the supernatant was removed.

10. The pellet was washed with 1000 μl of 75% ethanol (diluted with RNase-free water). Then, the samples were centrifuged at 7500 rpm at 4˚C for 2 min.

11. The supernatant was carefully removed and the washing process was repeated once more (step 10). The supernatant was completely removed and air-dried for 10 min

12. The pallet was dissolved with 20 μl of RNase-free water. The samples were placed in the water bath at 45˚C for 5 minutes and immediately held in ice for 2 minutes.

13. The extracted RNA was stored at -80˚C prior to characterisation analysis. **Fig 1** illustrates the QIAzol (Method A) method and the QIAzol + glycogen (Method B) method.

QIAzol + glycogen (Method B).

1. Use a similar method with Method A but with the addition of 2 μl of RNA-grade glycogen during the RNA precipitation process.

Qiagen Micro Kit (Method C).

1. Ten (10) and thirty (30) individuals of female *M. micrura* were transferred into a microcentrifuge tube.

2. The samples were homogenised in of RLT buffer (350 μl) and β mercaptoethanol (4 μl) by using a similar procedure to that of the previous method.

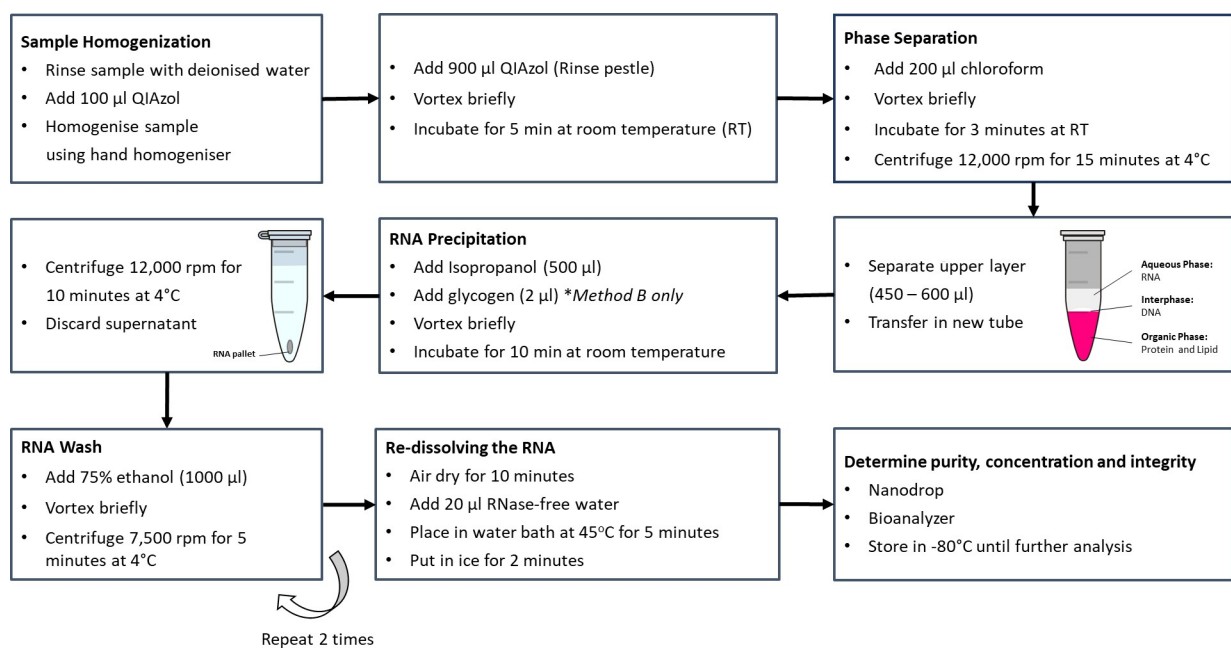

**Fig 1. Method of extraction by using QIAzol (Method A) and QIAzol + glycogen (Method B).**

3. The Qiagen Micro Kit was used according to the manufacturer's protocol.

4. To avoid RNA degradation, all centrifugations were carried out at 4°C and samples were kept on ice during the entire procedure

5. The RNA was eluted in RNase-free water (20 μl).

6. The extracted RNA was stored at -80°C prior to characterisation analysis. **Fig 2** illustrates the Qiagen Micro Kit (method C) method and the Qiagen Micro Kit + glycogen (Method D) method.

Qiagen Micro Kit + glycogen (Method D).

1. Use a similar method with Method C but with the addition of 2 μl of RNA-grade glycogen (Thermo Fisher Scientific, MA, United States) during the RNA precipitation process.

## RNA characterisation

The quantity (RNA yield) and quality (A260/230 and A260/280 ratios) of total RNA were measured by the Genova Nano spectrophotometer Jenway (OSA, UK). The integrity of RNA (RIN value) was measured by Agilent 2100 Bioanalyzer Agilent Technologies (Waldbronn, Germany). The bioanalyzer instrument used the Agilent RNA 6000 Pico Kit and was supplied by Agilent Technologies (Waldbronn, Germany).

## First-strand complementary DNA (cDNA) Synthesis and Reverse-Transcription Polymerase Chain Reaction (RT-PCR)

Two (2) replicate of RNA extraction samples from the best extraction method were used for cDNA synthesis and subsequent RT-PCR using primers based on previous literature [50].

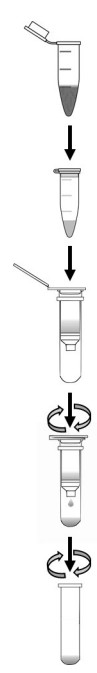

**Lyse and homogenise**
- Add 350 μl Buffer RLT. (Add 10 μl β-Mercaptoethanol (β-ME) per 1 ml Buffer RLT)
- Add 5 μl of Carrier RNA working solution (4 ng/μl)
- Disruption and homogenization using hand homogeniser
- Centrifuge the lysate for 3 min at full speed. Carefully transfer the supernatant to a new microcentrifuge tube by pipetting.

**Add ethanol**
- Add 1 volume (usually 350 μl) of 70% ethanol to the lysate and mix well by pipetting. Do not centrifuge. Add glycogen (2 μl) *Method D only.

**Bind total RNA and digest DNA**
- Transfer the sample, including any precipitate that may have formed, to an RNeasy MinElute spin column placed in a 2 ml collection tube. Incubate in room temperature for 2 minutes. Centrifuge for 15 s at ≥8000 x g (≥10,000 rpm). Discard the flowthrough.
- Add 350 μl Buffer RW1 to the RNeasy MinElute spin column. Centrifuge for 15 s at ≥8000 x g (≥10,000 rpm). Discard the flowthrough.
- Add the DNase I incubation mix (80 μl) directly to the RNeasy MinElute spin column membrane, and place on the benchtop (20–30°C) for 15 min.
- Add 350 μl Buffer RW1 to the RNeasy MinElute spin column. Centrifuge for 15 s at ≥8000 x g (≥10,000 rpm). Discard the flowthrough.

**Wash**
- Place the RNeasy MinElute spin column in a new 2 ml collection tube (supplied). Add 500 μl Buffer RPE to the spin column. Incubate in room temperature for 2 minutes and centrifuge for 15 s at ≥8000 x g (≥10,000 rpm).
- Add 500 μl of 80% ethanol to the RNeasy MinElute spin column. Incubate in room temperature for 2 minutes. Centrifuge for 2 min at ≥8000 x g (≥10,000 rpm). Discard the flow-through and collection tube.

**Elute**
- Place the RNeasy MinElute spin column in a new 2 ml collection tube (supplied). Open the lid of the spin column, and centrifuge at full speed for 5 min. Discard the flow-through and collection tube.
- Place the RNeasy MinElute spin column in a new 1.5 ml collection tube (supplied). Add 20 μl RNase-free water directly to the center of the spin column membrane. Incubate in room temperature for 2 minutes. Centrifuge for 1 min at full speed to elute the RNA.
- Store in -80°C until further analysis.

**Fig 2. Method of extraction by using Qiagen Micro Kit (method C) and Qiagen Micro Kit + glycogen (Method D).**

Tetro cDNA Synthesis Kit (Bioline, USA) was utilised to synthesis complementary DNA (cDNA) by mixing 2 μg/μl of total RNA, 8 μl of Reverse Transcriptase Buffer, 2 μl of 10 mM dNTP mix, 2 μl of Oligo (dT)18 Primer, 2 μl of Ribosafe RNase Inhibitor, 2 μl of Tetro Reverse Transcriptase and RNase free Water to make up to 40 μl. The reaction mixture was homogenised by pipetting gently and incubated inside T100™ thermal cycler (Bio-Rad, USA) at 45°C for 30 minutes. This was followed by an incubation at 85°C for 5 minutes to stop the reverse transcriptase reaction and the holding temperature was finally held at 4°C. The cDNA was kept at -20°C before RT-PCR analysis.

Two (2) genes (**Table 2**) will be validated using SensiFAST™ SYBR No-ROX Kit (Bioline, USA). 0.3 μg of cDNA template will be mixed with 0.4 μl of 10 μM forward qPCR primer, 0.4 μl of 10 μM reverse qPCRprimer, 10 μl of 2 x SensiFast SYBR No-ROX Mix (containing 10 μM of dNTP mixture, 3 mM of $MgCl_2$, SYBR® Green I dye, Taq Polymerase buffer, Taq DNA Polymerase, stabilisers and enhancers) and 8.2 μl of PCR grade distilled water. Each mixture will be pipetted into a 0.2 ml PCR tube, vortexed and centrifuged for a short spin prior to placing it into the Rotor-Gene Q thermocycler (QIAGEN, Germany). Two step RT-PCR will be performed with the cycling conditions consisting of an initial denaturation step at 95°C for 2 minutes, 40 cycles of denaturation (95°C for 5 seconds) and 65°C of annealing reaction.

**Table 2. Genes and primers used in the present study for *Moina micrura* [50].**

| Gene | Function | Sequences |
|---|---|---|
| Alpha Tubulin, aTub | Make up the cell's structural framework | Forward: TGGAGGTGGTGACGACT<br>Reverse: CCAAGTCGACAAAGACAGCA |
| Actin, Act | Makes up the structural framework inside cells. | Forward: CCACACTGTCCCCATTTATGAA<br>Reverse: CGCGACCAGCCAAATCC |

## Statistical analysis

Data analysis was performed using statistical software IBM SPSS (Version 25). One-way analysis of variance (ANOVA) and Tukey's post hoc pair-wise comparison test were employed to assess the effect of different RNA extraction methods on RNA quantity, quality (A260/280 and A260/230 ratios) and integrity (RIN values).

# Results and discussion

## RNA quantity

The efficacy of total RNA extraction methods should be validated by examining the quantity, quality and integrity of RNA to avoid repeated experiments, utilisation of toxic reagents, labour costs and instrumentation [51]. The comparison of the quantity (ng/individual), purity (A260/230 and A260/280 ratios) and integrity (RIN values) of the total RNA extracted from zooplankton using different extraction and preservation methods (**Table 3** and **S1 File**). The current study utilised different sample pools of 10 individuals and 30 individuals for both extraction methods. For both extraction methods, RNA concentration was higher in sample pools of 30 individuals than in sample pools of 10 individuals. A similar result was observed by Roy et. al. (2020), suggesting that when the initial sample size or mass is increased, the RNA concentration will also increase. Moreover, the Individual contents of RNA in *Artemia salina* were increased exponentially with the life stages and number of incubation days as observed

**Table 3. The comparison of quantity (ng/individual), purity (A260/230 and A260/280 ratios) and integrity (RIN values), of total RNA extracted from zooplankton using different extraction and preservation methods.**

| Extraction method | Preservation method | Species | N | n | RNA concentration (ng/μl) | A260/230 | A260/280 | Gel profile 18s | Gel profile 28s | RIN | References |
|---|---|---|---|---|---|---|---|---|---|---|---|
| | | | | | | | | 18s | 28s | | |
| Phenol-chloroform extraction | | | | | | | | | | | |
| TRIzol | None | *Calanus helgolandicus* | 5–15 | 9 | 54.62 ± 24.9 | 1.27 ± 0.48 | 1.81 ± 0.13 | + | - | 3.90 ± 1.13 | [51] |
| QIAzol | None | *Moina micrura* | 10 | 6 | 35.61 ± 20.9 | 1.77 ± 0.24 | 1.67 ± 0.28 | + | - | 5.87 ± 0.18 | This study |
| QIAzol | None | *Moina micrura* | 30 | 4 | 105.16 ± 16.0 | 1.68 ± 0.04 | 1.53 ± 0.11 | + | - | 5.70 ± 0.32 | This study |
| QIAzol + glycogen | None | *Moina micrura* | 10 | 6 | 113.71 ± 27.1 | 1.84 ± 0.05 | 1.86 ± 0.26 | + | - | 6.25 ± 0.10 | This study |
| QIAzol + glycogen | None | *Moina micrura* | 30 | 4 | 352.59 ± 22.2 | 1.66 ± 0.13 | 1.43 ± 0.20 | + | - | 6.10 ± 0.22 | This study |
| Column-based kit | | | | | | | | | | | |
| Qiagen mini kit | TRIzol | *Acartia hudsonica* | 50 | 70 | 5.34 ± 0.6 | NA | NA | NA | NA | NA | [54] |
| Aurum Total RNA Mini Kit | TRIzol reagent | *Calanus helgolandicus* | 5–15 | 16 | 7.25 ± 4.3 | 1.49 ± 0.66 | 1.99 ± 0.21 | + | - | 3.93 ± 1.19 | [51] |
| Aurum Total RNA Mini Kit | RNAlater | *Calanus helgolandicus* | 5–15 | 10 | 8.57 ± 2.0 | 1.58 ± 0.61 | 2.06 ± 0.08 | + | + | 9.43 ± 0.53 | [51] |
| Qiagen micro kit | TRIzol reagent | *Calanus helgolandicus* | 5–15 | 8 | 53.25 ± 14.4 | 2.47 ± 0.26 | 2.05 ± 0.04 | + | - | NA | [51] |
| Qiagen micro kit | RNAlater | *Calanus helgolandicus* | 5–15 | 12 | 57.69 ± 12.0 | 1.94 ± 0.40 | 2.02 ± 0.04 | + | + | 9.90 ± 0.14 | [51] |
| Qiagen micro kit | None | *Moina micrura* | 10 | 4 | 26.90 ± 6.9 | 1.95 ± 0.15 | 1.85 ± 0.09 | + | + | 7.20 ± 0.16 | This study |
| Qiagen micro kit | None | *Moina micrura* | 30 | 4 | 77.80 ± 3.4 | 1.88 ± 0.08 | 1.65 ± 0.12 | + | + | 6.70 ± 0.58 | This study |
| Qiagen micro kit + glycogen | None | *Moina micrura* | 10 | 4 | 17.52 ± 5.8 | 1.59 ± 0.17 | 1.55 ± 0.07 | + | + | 6.73 ± 0.27 | This study |
| Qiagen micro kit + glycogen | None | *Moina micrura* | 30 | 4 | 26.88 ± 6.4 | 1.58 ± 0.09 | 1.37 ± 0.14 | + | + | 6.15 ± 0.15 | This study |

Values represent the mean ± standard deviation. *NA* no values were assigned. *N* number of individuals per extraction. *n* number of replicates.

by Kobari et. al. (2017). The study also suggested that samples of early life stages of zooplankton do not contain RNA in amounts that are sufficient for any molecular study. Thus, extra individuals in the sample pools are needed during RNA extraction from zooplankton samples in the early life stages [52]. For this reason, phenol-chloroform method extraction is the best method of RNA extraction during the early life stages due to the fact that it can yield 2.4 to 93 times more RNA than the column-based kit method [39]. Furthermore, RNA concentration plateaus in the middle to late life stages (22–27 days of incubation). Thus, to ensure a sufficient concentration of RNA, the appropriate life stage for zooplankton to be taken as a sample is from 9 days onwards [52].

Overall, the phenol-chloroform method showed a higher quantity of RNA compared to the column-based kit method. Spectrophotometric analysis indicates that the use of glycogen increased the RNA yield in the phenol-chloroform method. In the sample pools consisting of 10 individuals, the addition of glycogen successfully increased the RNA yield by three (3)fold from $35.61 \pm 20.91$ ng/µl to $113.71 \pm 27.10$ ng/µl. However, numerous precautions must be taken during extraction when using this method because several contaminants such as phenol, chloroform, salt and guanidine may persist in samples. In this study, a micro kit is used as the representative of a column-based kit instead of a mini kit because it is suitable for purification of total RNA using samples as small as 5 mg for animal tissues or $5 \times 10^5$ cells for animal cells [53]. Study by Asai *et. al.* (2015) shows the RNA extracted from zooplankton by using the micro kit yielded more RNA compared to a mini kit, with RNA concentrations of $57.69 \pm 12.0$ and $8.57 \pm 2.0$, respectively. Additionally, the column-based micro kit method saves time, can be conducted rapidly and possesses easier protocol compared to the phenol-chloroform extraction method [39]. However, the addition of glycogen in the column-based kit caused a significant decrease in RNA yield from $26.90 \pm 6.89$ ng/µl to $17.52 \pm 5.84$ ng/µl. It is suggested that the mix between glycogen and lysate in the spin columns will interfere with the dynamics of column extraction. Subsequently, whenever the number of samples is high or samples are taken during the middle and late life stages, the column-based kit is the best extraction method, as it successfully eliminates residual contaminants, thus increasing the RNA quality and integrity.

## RNA quality and integrity

The total RNA extracted with the phenol-chloroform from the sample taken in the pools consisting of 10 individuals method had higher purity (A260/230 and A260/280 ratios) with the addition of glycogen during the extraction process. However, the column-based kit showed a decrease in A260/230 and A260/280 values when glycogen was added during the extraction process. In this study, the Qiagen micro kit without the addition of glycogen resulted in the best A260/230 and A260/280 ratio values, or $1.95 \pm 0.15$ and $1.85 \pm 0.09$, respectively. Nevertheless, in the sample pools consisting of 30 individuals, the addition of individuals into the pools decreased the A260/230 and A260/280 values for both extraction methods. This is because well-known contaminants such as ribonucleases (RNases) converge more strongly in the larger pooled samples, thus extensively digesting the RNA in the sample [54]. To prevent this phenomenon from occurring, multiple washing steps are required in extraction methods to eliminate the contaminants in the samples. Subsequently, rapid extraction methods and avoidance of environmental heat are also essential to ensure high-quality RNA in the samples [55]. The quality of the RNA extracted, with a reading of A260/230 and an A260/280 ratio between 1.8–2.0, indicated that the extracted RNA was free from contaminants and suitable for downstream applications [54].

The current study indicates that the addition of individuals in the sample pools reduces RNA integrity for both extraction methods. This result is due to numerous contaminants

such as RNases becoming more concentrated in the pooled samples [54]. Thus, the current study established 10 individuals per sample pool as the optimised sample size for both extraction methods. **Fig 3** shows the bioanalyzer assessment of the RNA extracted from 10 individuals of female *M. micrura*. **Fig 3B** shows glycogen did not cause 18S rRNA and 28S rRNA to peak, indicating that glycogen did not interfere with any of the proposed methods. The heat denaturing step, which stored RNA at 70˚C for 2 min before loading it onto the chip, was excluded in this method. The denaturing step is recommended in the Agilent 2100 Bioanalyzer manufacturer's instructions. However, this step causes the 28S rRNA peak to disappear from the electropherograms. During the heat denaturation step, the two fragments of 28S rRNA separate and migrate with the 18S rRNA. This phenomenon was associated with the 'hidden break' present in protostome 28S rRNA [56]. A similar suggestion was made by Asai et al. (2015) to skip the heat denaturation step when assessing RNA integrity with zooplankton species. Additionally, RNA integrity is one of the most vital factors in the effectiveness of RNA extraction, since a low integrity value could intensely influence downstream analysis [57,58].

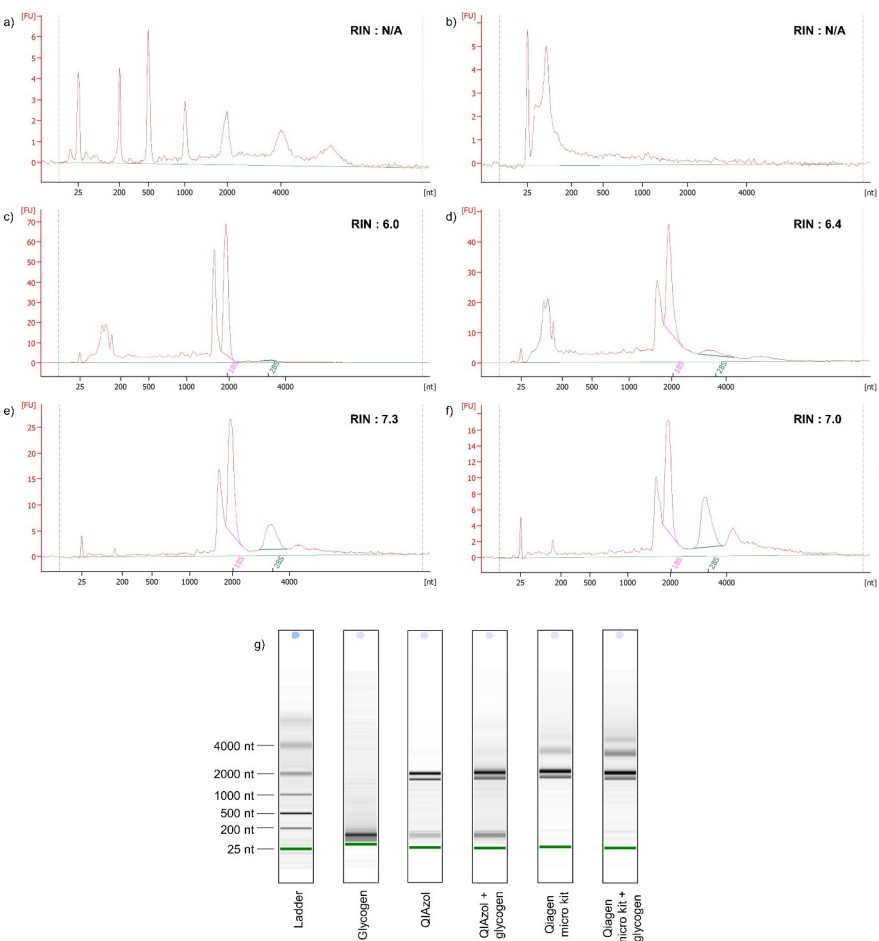

**Fig 3. The bioanalyzer assessment of total RNA from ten (10) female *M. micrura* extracted using different RNA extraction methods with the agilent RNA 6000 pico kit on the agilent 2100 bioanalyzer.** X-axis units in nt (Nucleotides); Y-axis units in FU (Fluorescence Units); N/A (Not Applicable) a) RNA ladder, b) glycogen, c) QIAzol method, d) QIAzol + glycogen method, e) Qiagen Micro Kit method, f) Qiagen Micro Kit + glycogen method, and g) Electrophoretic gel for each sample.

Glycogen effectively increased the RNA integrity of the phenol-chloroform method from $5.87 \pm 0.18$ to $6.25 \pm 0.10$, However, glycogen has undesirable effects on the column-based kit as it was shown to decrease RNA integrity from $7.20 \pm 0.16$ to $6.73 \pm 0.27$. Total RNA from *M. micrura* extracted with the phenol-chloroform method showed a single peak corresponding to the 18S rRNA subunit, an absence of a 28S rRNA peak and a small amount of small RNA occurring between 25 nt and 200 nt. Subsequently, the column-based kit showed a clear 28S rRNA peak and a tiny amount of small RNA appearing between 25 nt and 200 nt. Additionally, the RIN values obtained by using column-based kit methods were significantly higher than those achieved with the phenol-chloroform method. The low RIN values, particularly in the phenol-chloroform method, were associated with the absence of 28S rRNA. The dissociation of the 28S rRNA was caused by the denaturation process of hydrogen bond breaking during the RNA extraction [51]. A similar discovery related to the denaturing effect of the lysis reagent on the 28S rRNA was reported in molluscs, insects and other zooplankton species [44]. A previous extraction study shows that the dissociation of 28S RNA caused by using the phenol-chloroform extraction method causes the RIN value to drop to $3.90 \pm 1.13$ [51]. As a result, the phenol-chloroform method is suggested to be unsuitable for extracting RNA from these organisms. Several studies showed that the extracted RNA in a RIN value of 6–10, indicating RNA from human and animal tissues that is high quality and non-degraded, and thus suitable for downstream applications [59,60]. This study showed that the column-based kit extraction is the most reliable method to extract RNA from freshwater Cladocerans species, specifically *M. micrura*. The effects of glycogen on RNA quantity, quality (A260/230 and A260/280 ratios) and integrity (RIN values) in both phenol-chloroform and column-based kit methods (**Fig 4**).

The Qiagen Micro Kit (Method C) was the most optimised method and surpassed the minimum requirement of RNA quality control. Thus, the RNA extracted from this method was

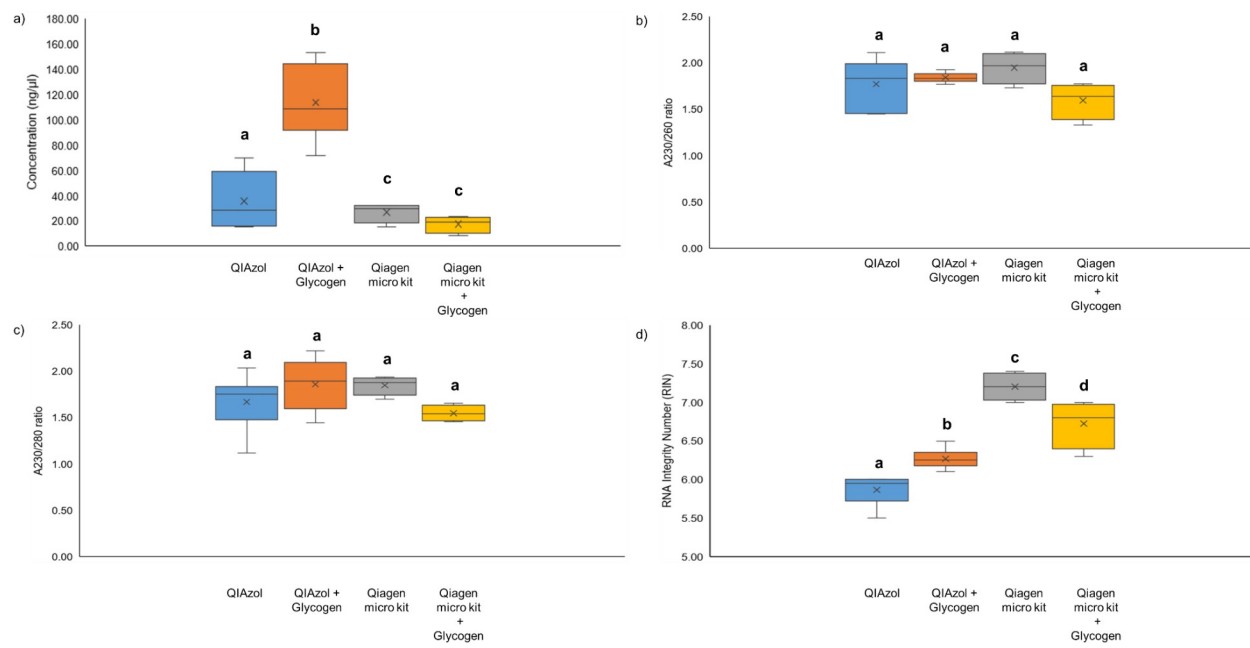

**Fig 4. The effect of glycogen on RNA quantity, quality (A260/230 and A260/280 ratios) and integrity (RIN values).** Box plots illustrate a) total yield of RNA extracted, b) A230/260 ratios of RNA extracted, c) A230/280 ratios of RNA extracted and d) RIN values of RNA extracted. Significant differences were analysed by ANOVA (Tukey's post hoc test; $\rho < 0.05$) and are expressed as different letters. The results are based on the RNA extraction from the sample pools of ten (10) female *M. micrura*.

used for cDNA synthesis and subsequent RT-PCR analysis. Alpha tubulin and actin genes were employed as housekeeping genes in this study. Both genes have been identified as cladocerans' cellular maintenance genes that regulate basic and ubiquitous cellular functions [50]. Thus, the genes were ideal candidates for serving as internal controls for gene expression analysis towards cladocerans. Moreover, the consistent expression of a housekeeping gene is required in any RT-PCR analysis since it is crucial for in the interpretation of the final result [61]. In all samples, the RT-PCR amplification cycle graphs showed that both alpha tubulin and actin genes exponential increase started at 33–35 cycles and the cycle threshold (Ct) values were between 32.64–33.48, respectively (**S2 File**). The stable reference genes identified in this study indicates current extraction method produces useful genetic material for testing the hypotheses involving genetic expression assays.

Another factor that can influence the quality and integrity of RNA used for downstream analysis besides the extraction method is the preservation of RNA samples. Previous studies have utilised several preservation methods, including preservation in RNAlater and storage at -80˚C [44,51,54,62–67], preservation in lysis reagent and storage at -80˚C [51,54,62,64], preservation in ethanol and storage at -80˚C [38,68] and immersion in liquid nitrogen and storage at -80˚C [62,63,65]. Preservation in formalin or 95–100% ethanol and flash freezing by using liquid nitrogen is a conventional method and most regularly used for bulk fixation of zooplankton samples [69,70]. However, this method does not preserve the integrity of proteins and degrades the genomic RNA. Moderate nucleic acid degradation in ethanol-preserved samples of *Parvocalanus crassirostris* after a month of preservation was observed by Goetze and Jungbluth (2013). The conventional method requires toxic chemicals and relies on liquid nitrogen, which is expensive. TRIzol as a preservation reagent for zooplankton, as no significant RNA degradation ($p > 0.05$) was detected in *Acartia hudsonica* preserved in TRIzol at 4˚C for up to two weeks, or −80˚C for two years as suggested by Zhang et. al. (2013). A similar study indicated RNAlater was not a suitable preservation reagent for *Acartia hudsonica* because no RNA was detectable in 6 out of 24 samples during the extraction process after preservation in RNAlater. The study indicated RNAlater fixation is not suitable for the preservation of individual small-sized zooplankton. Thus, the number of zooplankton samples preserved in RNAlater should be increased to reduce RNA loss during the preservation process [54]. Different observations were obtained by Gorokhova (2005), who revealed that RNAlater was capable of preserving RNA from an *Artemia* spp. sample for about 1 month at room temperature and 4 months at 5˚C. Subsequently, the study indicates that -80˚C is the best temperature for sample storage because no significant RNA degradation was observed until after 8 months. Accordingly, if the sample requires longer storage time, the temperature during storage must be kept as low as possible to ensure the RNA in the sample remains intact and is not degraded. Previous study by Asai et. al. (2015) also proposed RNAlater as a preservation reagent if liquid nitrogen is not available for flash freezing of zooplankton samples. A similar study indicated the integrity of Calanus helgolandicus was increased by 2.5-fold when it was preserved with RNAlater, from 3.90 ± 1.13 to 9.9 ± 0.14. Additionally, the effects of RNAlater on zooplankton depend on the exoskeleton structure of zooplankton species. A thin and non-calcified cuticle exoskeleton will increase the effectiveness of RNA preservation, as the RNAlater can easily penetrate the tissue, particularly before the samples are refrigerated. Preservation methods are essential to maintain the integrity of RNA from samples, thus determining the success of downstream analysis. Therefore, further studies by using several preservation methods before the RNA extraction are recommended to increase the integrity of RNA from several zooplankton samples.

Furthermore, the occurrence of RNases in both the sample and the environment causes the breakdown of RNA into smaller components [56]. The hydroxyl (-OH) group and diatomic

carbon (C2) group in RNA chemical structure play a crucial role in the nonenzymatic degradation process [71]. Degraded RNA will not become amplified at the same level as cDNA derived from intact RNA during gene expression analysis. As RNA becomes degraded, quantitative expression levels determined by RT-PCR decrease and may cause inaccurate and unreliable gene expression assessment in the samples. RNA sequencing (RNA-seq) is an established high-throughput sequencing assay that applies next-generation sequencing (NGS) to determine any changes in the cellular transcriptome [72,73]. The transcript level of various genes changes with even slight differences in RNA integrity as observed by Reiman et. al. (2017). The highly degraded RNA did not yield enough sequencing library data; thus, the diversity and the coverage of transcript level values were significantly understated [65]. Samples with RIN values below 2.2 showed significant differences in gene expression profiles, producing only 30 million sequenced reads compared to 50 million sequenced reads of high-quality RNA [65]. However, data normalisation techniques can be implemented to increase the accuracy of gene expression assessments in RNA-seq analysis for degraded samples [74]. Data normalisation during RNA-seq analysis could determine several expressed genes, even when RNA samples with RIN values as low as 4 were included as examined by Romero et. al. (2014).

## Strengths, Weaknesses, Opportunities, and Threats (SWOT) analysis

Worldwide communities must integrate all available ecotoxicological data, especially at molecular levels, by providing more detailed mechanistic information. The Organisation

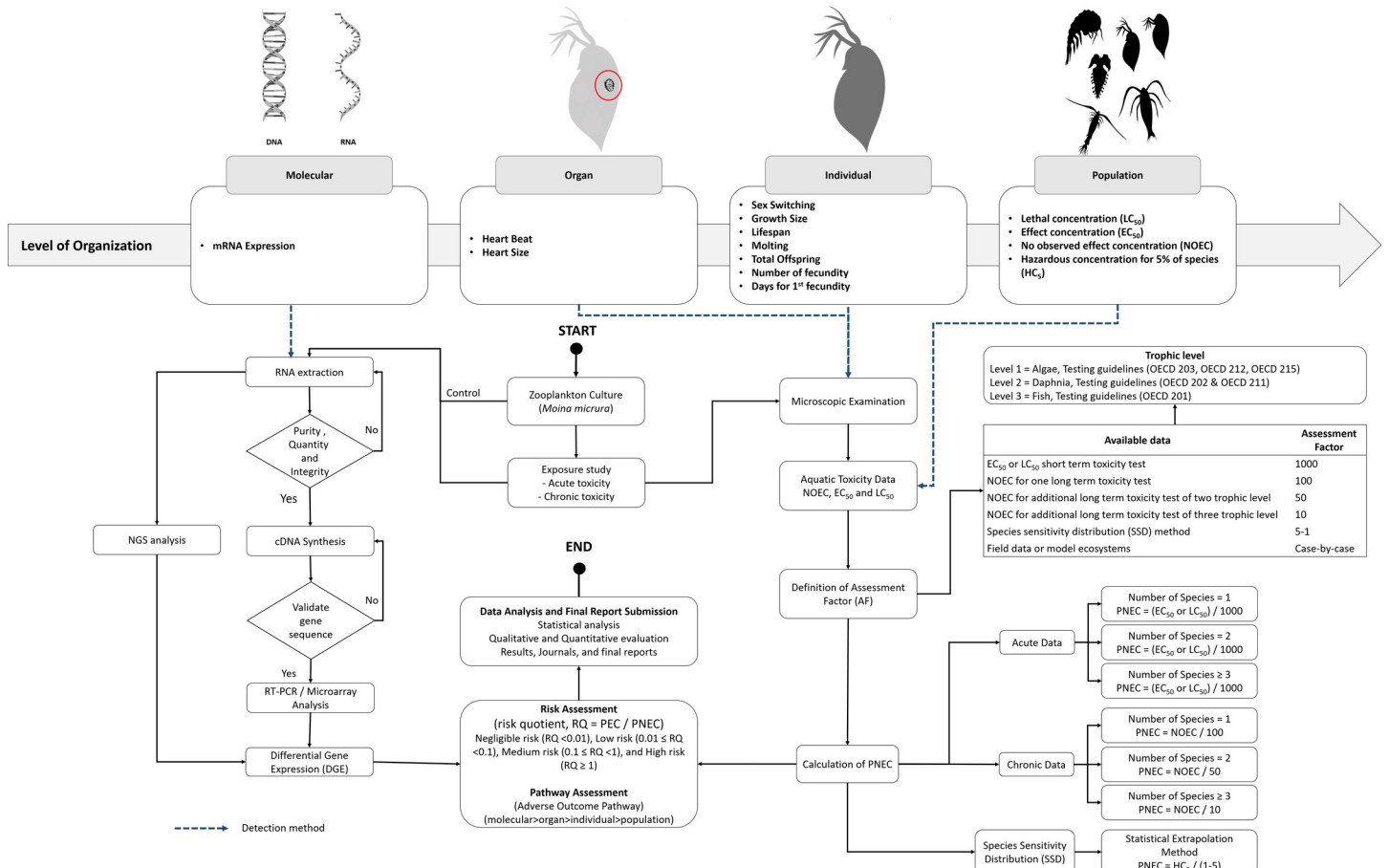

**Fig 5.** a) Key apical endpoints in various levels of organisation and b) workflow of Cladocerans bioindicator in risk and pathway assessment.

for Economic Co-operation and Development (OECD) established the adverse outcome pathway (AOP) framework to compile, incorporate and integrate ecotoxicological data through pathway assessment to clarify adverse outcomes. The framework examines the

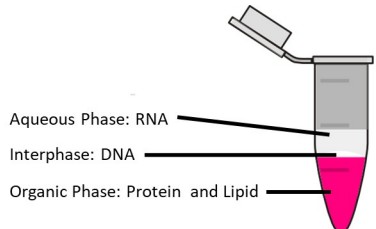

# Phenol-chloroform extraction

| STRENGTH | WEAKNESS | OPPORTUNITIES | THREATS |
|---|---|---|---|
| • Very efficient lysis<br>• High yield<br>• Less expensive | • Harmful chemicals (chloroform)<br>• Longer procedure<br>• Possible residue contaminants (phenol and salt)<br>• Needs excellent pipetting skills | • Ability to combine with nucleic acid carrier (glycogen)<br>• Ability to integrate with purification kit | • Not suitable for NGS and other downstream application towards zooplankton samples |
| • Fast procedure<br>• Easier protocol of RNA extraction<br>• Purer RNA | • Lower yield<br>• Harmful chemicals (β-mercaptoethanol)<br>• Expensive | • Wide range of RNA extraction kits covering many sample types and throughput needs<br>• Suitable for NGS and other downstream application towards zooplankton samples | • Short expiration date |

# Column-based kit

**Fig 6. SWOT analysis of the phenol-chloroform extraction method and the column-based kit for the RNA extraction of zooplankton samples.**

effects of chemicals through several levels of biological organisation (e.g., neuronal, cells, tissue, organ and individual) leading to adverse outcomes that are important to chemical and risk assessment [75]. **Fig 5** shows a) key apical endpoints in several organisation levels and b) workflow at the molecular level for risk and pathway assessment. The figure shows the critical molecular assessment as a core interaction that causes adverse outcomes to the organisms. As more ecotoxicological data at the molecular level is recorded, it is possible to predict and evaluate potential adverse outcomes based on toxicity mechanistic understanding [76]. OECD established AOP Knowledge Base (AOP-KB) as the central repository for accelerating AOP applications (https:/aopkb.oecd.org/). AOP-KB consists of AOP-Wiki, Effectopedia, and AOP Xplorer for gathering mechanistic information and evaluating existing AOPs. The framework provides a means of collecting, organising and integrating data from numerous sources to allow accurate evaluations for chemical risk assessment, hazard projection and regulatory decision making [77]. Comparison assessment between vertebrate and invertebrate species is essential to determine the relationships in both species. The latest study shows several similar reactions across both species at the molecular levels of biological organisation, indicating potentially numerous overlapping responses to specific stressors. Unfortunately, only a fraction of molecular assessment data is reported on invertebrate species, especially zooplankton. Subsequently, the comparative analysis of two commonly used RNA extraction approaches in this study will support studies on cladocerans as non-model organisms for molecular assessment. A Strengths, Weaknesses, Opportunities, and Threats (SWOT) analysis of the phenol-chloroform extraction method and the column-based kit for the RNA extraction of zooplankton samples (**Fig 6**). Although the column-based kit is more expensive than the phenol-chloroform extraction method, the column-based kit offers high-quality extracted RNA and a rapid method that is more suitable for downstream application with zooplankton samples.

## Conclusions

RNA quality control is an essential step when evaluating gene expression profiles for downstream applications. In this study, a comprehensive assessment of RNA extracted from *M. micrura* was conducted using phenol-chloroform and column-based extraction methods. This study also showed the usefulness of a well known inert carrier, glycogen, in enhancing the quantity, quality and integrity of RNA, albeit only for a specific extraction method, namely the phenol-chloroform method. However, the phenol-chloroform method is not compatible with *M. micrura* or other zooplankton, insects and molluscs species because the lysis reagent used in this method will eventually denature 28S rRNA. On the other hand, the incorporation of glycogen is not compatible with the column-based kit method due to its interference with the dynamics of column extraction. However, the column-based kit method surpassed the minimum requirement of RNA quality control for downstream applications such as reverse transcription polymerase chain reaction (RT-PCR) and next-generation sequencing (NGS).

## Supporting information

**S1 File. Quantity, purity and integrity of the total RNA.**
(XLSX)

**S2 File. RT-PCR cycle graphs and bioanalyzer raw data.**
(DOCX)

## Acknowledgments

The Authors would like to acknowledge Aquatic Animal Health and Therapeutics Laboratory (AquaHealth), Universiti Putra Malaysia for laboratory equipments. First author also would like to thank Aisamuddin Ardi bin Zainal Abidin for his kind assistances during the course of this project.

## Author Contributions

**Conceptualization:** Muhammad Raznisyafiq Razak, Ahmad Zaharin Aris.

**Data curation:** Muhammad Raznisyafiq Razak.

**Formal analysis:** Muhammad Raznisyafiq Razak, Ahmad Zaharin Aris.

**Funding acquisition:** Ahmad Zaharin Aris.

**Methodology:** Muhammad Raznisyafiq Razak, Zetty Norhana Balia Yusof.

**Software:** Muhammad Raznisyafiq Razak.

**Supervision:** Ahmad Zaharin Aris.

**Visualization:** Muhammad Raznisyafiq Razak.

**Writing – original draft:** Muhammad Raznisyafiq Razak.

**Writing – review & editing:** Muhammad Raznisyafiq Razak, Ahmad Zaharin Aris, Fatimah Md Yusoff, Zetty Norhana Balia Yusof, Sang Don Kim, Kyoung Woong Kim.

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
