## [Decision Letter · Decision Letter 0]

11 Nov 2021

PONE-D-21-28380Assessment of RNA extraction protocols from CladoceransPLOS ONE

Dear Dr. Aris,

Thank you for submitting your manuscript to PLOS ONE. After careful consideration, we feel that it has merit but does not fully meet PLOS ONE’s publication criteria as it currently stands. Therefore, we invite you to submit a revised version of the manuscript that addresses the points raised during the review process.

We look forward to receiving your revised manuscript.

Kind regards,

SSS Sarma

Academic Editor

PLOS ONE

Journal Requirements:

Additional Editor Comments (if provided):

I am sorry to provide you the reviewer's observations very late. This is not the case for Plos One which has a reputation of rapid and objective evaluation of every manuscript. One reviewer, after having agreed to review, did not respond even after 2 weeks passing the deadline. Anyhow, based on one reviewer, I noticed that your methods are flawed. I agree with her/him. Anyhow, I would like to give an opportunity to re-submit a revised version, by taking into account of all observations of the reviewer. To be fair to the authors, the revised version will be sent out to completely new set of reviewers.

Sincerely

Handling Editor

Reviewers' comments:

Reviewer's Responses to Questions

**Comments to the Author**

1. Is the manuscript technically sound, and do the data support the conclusions?

Reviewer #1: Yes

2. Has the statistical analysis been performed appropriately and rigorously? 

Reviewer #1: Yes

3. Have the authors made all data underlying the findings in their manuscript fully available?

Reviewer #1: Yes

4. Is the manuscript presented in an intelligible fashion and written in standard English?

Reviewer #1: Yes

5. Review Comments to the Author

Reviewer #1: The manuscript entitled “Assessment of RNA extraction protocols from Cladocerans” (PONE-D-21-28380) is well written and carefully designed, the results were treated appropriately, and the conclusions are supported by the experimental results. However the exprimental design focuses only on the standardization of two RNA extraction methods (the phenol-chloroform extraction method and a column-based kit) using a biological model (Moina micrura) that is of interest in ecological and toxicology studies. The authors, based on qualitative and quantitative indicators, concluded that the genetic material obtained could be used in downstream gene expression essays.

The standardization of existing protocols is the first phase of the experimental work that aims to test the molecular response of an organism to a specific condition. Therefore, I do not consider that this report in its current state deserves its publication in PLOS ONE journal. In my opinion, the authors could subject to Moina micrura to controlled conditions to prove that the extraction protocol produces useful genetic material to for testing hypotheses involving genetic expression essays.

6. PLOS authors have the option to publish the peer review history of their article (what does this mean?). If published, this will include your full peer review and any attached files.

Reviewer #1: No

---

## [Author Response · Author response to Decision Letter 0]

2 Jan 2022

Department of Environment

Faculty of Forestry and Environment

Universiti Putra Malaysia (UPM)

43400 UPM Serdang, Selangor, Malaysia

4th December 2021

SSS Sarma

Academic Editor

PLOS ONE

Dear Editor,

Title: Assessment of RNA extraction protocols from Cladocerans (PONE-D-21-28380).

We would like to express our gratitude for the useful comments suggested by the reviewers. The authors have made the necessary amendments to the manuscript accordingly. Hereby, we have provided the “Response to Reviewers” as follows:

EDITOR: General Comment

I am sorry to provide you the reviewer's observations very late. This is not the case for Plos One which has a reputation of rapid and objective evaluation of every manuscript. One reviewer, after having agreed to review, did not respond even after 2 weeks passing the deadline. Anyhow, based on one reviewer, I noticed that your methods are flawed. I agree with her/him. Anyhow, I would like to give an opportunity to re-submit a revised version, by taking into account of all observations of the reviewer. To be fair to the authors, the revised version will be sent out to completely new set of reviewers.

Answer to the editor comment: 

We would like to thank the Editor for the comment and for the opportunity to re-submit a revised version. The authors would like to sincerely appreciate all valuable comments and suggestions, which helped us to improve the quality of the article. The authors have added several improvements as suggested by the editor and reviewers in the manuscripts. The authors have amended material and methods section such as the addition of “M. micrura Culture Specifications”, “M. micrura Controlled Samples Collection” and “First-Strand Complementary DNA (cDNA) Synthesis and Reverse-Transcription Polymerase Chain Reaction (RT-PCR)” sub-sections. Moreover, authors added RT-PCR analysis for two (2) genes to prove current optimised extraction method (method C) produces useful genetic material for testing the hypotheses involving genetic expression assays (S1 Fig.). The additional discussion can be seen in the section below (general comments by reviewer #1).

Reviewer #1: General Comment

Reviewer #1: The manuscript entitled “Assessment of RNA extraction protocols from Cladocerans” (PONE-D-21-28380) is well written and carefully designed, the results were treated appropriately, and the conclusions are supported by the experimental results. However, the experimental design focuses only on the standardization of two RNA extraction methods (the phenol-chloroform extraction method and a column-based kit) using a biological model (Moina micrura) that is of interest in ecological and toxicology studies. The authors, based on qualitative and quantitative indicators, concluded that the genetic material obtained could be used in downstream gene expression essays. The standardization of existing protocols is the first phase of the experimental work that aims to test the molecular response of an organism to a specific condition. Therefore, I do not consider that this report in its current state deserves its publication in PLOS ONE journal. In my opinion, the authors could subject to Moina micrura to controlled conditions to prove that the extraction protocol produces useful genetic material to for testing hypotheses involving genetic expression essays.

Answer to the Reviewers comment: 

The experimental design was focuses on both standardisation of two RNA extraction methods and standardisation subject of Moina micrura to controlled conditions. Based on the suggestion by the reviewers, the authors have revised the information by adding “M. micrura Culture Specifications”, “M. micrura Controlled Samples Collection” and “First-Strand Complementary DNA (cDNA) Synthesis and Reverse-Transcription Polymerase Chain Reaction (RT-PCR)” sub-sections in the material and methods section. Moreover, authors added RT-PCR analysis for two (2) genes to prove current optimised extraction method (method C) produces useful genetic material for testing the hypotheses involving genetic expression assays (S1 Fig.). The authors have added several explanations regarding the subject of Moina micrura to controlled conditions in the revised manuscript as follow:

(Material and Methods Section)

M. micrura Culture Specifications 

Live samples of Cladocerans M. micrura were obtained from the culture in the Aquatic Animal Health and Therapeutics Laboratory (AquaHealth), Institute of Bioscience, Universiti Putra Malaysia. The culture was cultivated based on the International Organization for Standardization (ISO) method (ISO 6341:2012). The culture specifications were regulated to the light/dark cycle of 12: 12 h and 27 ± 1°C. The media was changed every two days with the daily supplementation of a green microalga species, Chlorella vulgaris (1.0 × 106~8 cells/ ml, 1 ml/day), as the food supply.

M. micrura Controlled Samples Collection

The standardization of sample collection to ensure the M. micrura in controlled condition throughout the RNA extraction process. Live M. micrura females were sorted from the culture sample under a Zeiss Axioskop 2 microscope (Zeiss, Germany). Females of M. micrura were collected similarly from the controlled culture prior to the RNA extraction. All experiments were performed in accordance with UK legislation under the Animals (Scientific Procedures) Act 1986 Amendment Regulations (SI 2012/3039).

First-Strand Complementary DNA (cDNA) Synthesis and Reverse-Transcription Polymerase Chain Reaction (RT-PCR).

Two (2) replicate of RNA extraction samples from the best extraction method were used for cDNA synthesis and subsequent RT-PCR using primers based on previous literature [35]. Tetro cDNA Synthesis Kit (Bioline, USA) was utilised to synthesis complementary DNA (cDNA) by mixing 2 µg/µl of total RNA, 8 µl of Reverse Transcriptase Buffer, 2 µl of 10 mM dNTP mix, 2 µl of Oligo (dT)18 Primer, 2 µl of Ribosafe RNase Inhibitor, 2 µl of Tetro Reverse Transcriptase and RNase free Water to make up to 40 µl. The reaction mixture was homogenised by pipetting gently and incubated inside T100™ thermal cycler (Bio-Rad, USA) at 45°C for 30 minutes. This was followed by an incubation at 85°C for 5 minutes to stop the reverse transcriptase reaction and the holding temperature was finally held at 4°C. The cDNA was kept at -20°C before RT-PCR analysis. 

Two (2) genes (Table 1) will be validated using SensiFAST™ SYBR No-ROX Kit (Bioline, USA). 0.3 µg of cDNA template will be mixed with 0.4 µl of 10 µM forward qPCR primer, 0.4 µl of 10 µM reverse qPCRprimer, 10 µl of 2 x SensiFast SYBR No-ROX Mix (containing 10 µM of dNTP mixture, 3 mM of MgCl2, SYBR® Green I dye, Taq Polymerase buffer, Taq DNA Polymerase, stabilisers and enhancers) and 8.2 µl of PCR grade distilled water. Each mixture will be pipetted into a 0.2 ml PCR tube, vortexed and centrifuged for a short spin prior to placing it into the Rotor-Gene Q thermocycler (QIAGEN, Germany). Two step RT-PCR will be performed with the cycling conditions consisting of an initial denaturation step at 95°C for 2 minutes, 40 cycles of denaturation (95°C for 5 seconds) and 65°C of annealing reaction.

Table 1. Genes and primers used in the present study for Moina micrura [35].

Gene Function Sequences

Alpha Tubulin, aTub Make up the cell's structural framework 

Forward: TGGAGGTGGTGACGACT

Reverse: CCAAGTCGACAAAGACAGCA

Actin, Act Makes up the structural framework inside cells. 

Forward: CCACACTGTCCCCATTTATGAA

Reverse: CGCGACCAGCCAAATCC

(Results and Discussion)

In this study, The Qiagen Micro Kit (Method C) was the most optimised method and surpassed the minimum requirement of RNA quality control. Thus, the RNA extracted from this method was used for cDNA synthesis and subsequent RT-PCR analysis. The RT-PCR amplification cycle graphs showed that for both genes in all samples, the exponential increase started at 33–35 cycles (i.e. Ct = 32.64 to 33.48) (S1 Fig.). The result indicates current extraction method produces useful genetic material for testing the hypotheses involving genetic expression assays.

(Supporting information)

Figure S1: Real time PCR amplification cycle graphs of the amplicons of two M. micrura genes. Two (2) replicate of RNA extraction samples from the best extraction method (Method C) were used for cDNA synthesis. a) Alpha tubulin; b) Actin.

---

## [Decision Letter · Decision Letter 1]

21 Feb 2022

Assessment of RNA extraction protocols from Cladocerans

PONE-D-21-28380R1

Dear Dr. Aris,

We’re pleased to inform you that your manuscript has been judged scientifically suitable for publication and will be formally accepted for publication once it meets all outstanding technical requirements.

Kind regards,

SSS Sarma

Academic Editor

PLOS ONE

Additional Editor Comments (optional):

One reviewer has asked for some minor corrections. These can be done at proof stage. The manuscript has been recommended for publication. However, the editor in chief has the final decision.

Sincerely

Handling Editor

Reviewers' comments:

Reviewer's Responses to Questions

**Comments to the Author**

1. If the authors have adequately addressed your comments raised in a previous round of review and you feel that this manuscript is now acceptable for publication, you may indicate that here to bypass the “Comments to the Author” section, enter your conflict of interest statement in the “Confidential to Editor” section, and submit your "Accept" recommendation.

Reviewer #1: All comments have been addressed

Reviewer #2: (No Response)

2. Is the manuscript technically sound, and do the data support the conclusions?

Reviewer #1: Yes

Reviewer #2: Partly

3. Has the statistical analysis been performed appropriately and rigorously? 

Reviewer #1: Yes

Reviewer #2: Yes

4. Have the authors made all data underlying the findings in their manuscript fully available?

Reviewer #1: Yes

Reviewer #2: Yes

5. Is the manuscript presented in an intelligible fashion and written in standard English?

Reviewer #1: Yes

Reviewer #2: Yes

6. Review Comments to the Author

Reviewer #1: (No Response)

Reviewer #2: Dear Authors

I highly appreciate the research concept and your hard efforts. I think outcomes could be more concrete and understandable to readers, especially the discussion parts. The research is focused on two RNA extraction methods, which I think it's acceptable. However, please reconstruct the abstract and discussion section then it will be more convincing to readers. I cordially suggest accumulating some more literature for supporting your methods and discussion.

7. PLOS authors have the option to publish the peer review history of their article (what does this mean?). If published, this will include your full peer review and any attached files.

Reviewer #1: No

Reviewer #2: No

---

## [Editor Report · Acceptance letter]

18 Apr 2022

PONE-D-21-28380R1 

Assessment of RNA extraction protocols from Cladocerans 

Dear Dr. Aris:

I'm pleased to inform you that your manuscript has been deemed suitable for publication in PLOS ONE. Congratulations! Your manuscript is now with our production department. 

Kind regards, 

on behalf of

Professor SSS Sarma 

Academic Editor

PLOS ONE